# Fabricating Graphene Oxide/h-BN Metal Insulator Semiconductor Diodes by Nanosecond Laser Irradiation

**DOI:** 10.3390/nano12152718

**Published:** 2022-08-08

**Authors:** Siddharth Gupta, Pratik Joshi, Ritesh Sachan, Jagdish Narayan

**Affiliations:** 1Department of Materials Science and Engineering, Centennial Campus, North Carolina State University, Raleigh, NC 27695, USA; 2Intel Corporation, Rolner Acres Campus 3, Hillsboro, OR 97124, USA; 3School of Mechanical and Aerospace Engineering, Oklahoma State University, Stillwater, OK 74078, USA

**Keywords:** Raman spectroscopy, h-BN, 2D materials, laser irradiation, graphene, molecular dynamics, Fowler-Nordheim tunneling, transmission electron microscopy

## Abstract

To employ graphene’s rapid conduction in 2D devices, a heterostructure with a broad bandgap dielectric that is free of traps is required. Within this paradigm, h-BN is a good candidate because of its graphene-like structure and ultrawide bandgap. We show how to make such a heterostructure by irradiating alternating layers of a-C and a-BN film with a nanosecond excimer laser, melting and zone-refining constituent layers in the process. With Raman spectroscopy and ToF-SIMS analyses, we demonstrate this localized zone-refining into phase-pure h-BN and rGO films with distinct Raman vibrational modes and SIMS profile flattening after laser irradiation. Furthermore, in comparing laser-irradiated rGO-Si MS and rGO/h-BN/Si MIS diodes, the MIS diodes exhibit an increased turn-on voltage (4.4 V) and low leakage current. The MIS diode I-V characteristics reveal direct tunneling conduction under low bias and Fowler-Nordheim tunneling in the high-voltage regime, turning the MIS diode ON with improved rectification and current flow. This study sheds light on the nonequilibrium approaches to engineering h-BN and graphene heterostructures for ultrathin field effect transistor device development.

## 1. Introduction

The drive to develop thin and light electronic devices, combined with device downscaling goals, has prompted researchers to focus their efforts on developing robust field effect transistors (FETs) that use thin insulating barriers that are complementary to graphene. The layered structure of h-BN, together with its 5.8 eV bandgap, makes it a promising candidate for this purpose [1]. It outperforms ordinary dielectric materials due to its improved oxidation resistance at temperatures up to 1700 K [2], minimal fixed charge at the interfaces, and dangling bonds [3]. Moving away from FETs, a graphene interface with a high-k dielectric, such as h-BN, is a promising candidate for applications such as electron field and deep UV emissions [4] and detection devices [5]. However, at the wafer scale, the complexity of fabricating large-area films and maintaining thickness uniformity presents substantial hurdles. These ultrathin materials must meet flatness, roughness, and disorder specifications to achieve long-term device integration. Impurities and voids in the transistors increase the device leakage current, lowering the transistor reliability and ON/OFF ratios.

Recent studies have demonstrated that graphene can form junctions with 3D or 2D semiconducting materials, which have rectifying characteristics and behave as excellent Schottky diodes. The main novelty of these devices is the tunable Schottky barrier height, a feature that makes the graphene–semiconductor junction an excellent platform for the study of the interface transport mechanisms and applications in photo-detection, high-speed communications, solar cells, chemical and biological sensing, etc. [6,7,8]. Bartolomeo et al. fabricated this device by transferring commercial graphene on a low-doped n-type Si substrate [9]. This diode was characterized by a Schottky barrier height of 0.52 eV at room temperature, an effective Richardson constant of 4 × 10^−5^ A cm^−2^ K^−2^, and an ideality factor of ≈ 3.6. In another study, the effect of temperature and light on the I-V and C-V characteristics of a graphene–silicon Schottky diode was studied. The device exhibited a photoresponsivity as high as 2.5 A/W [10].

Because of the precise control over the dopant direction during ion implantation, p-n junctions are more commonly used in device applications. Schottky diodes have substantially quicker switching times, which are required for high-power and high-speed devices. The MIS structure may improve the transportation properties of the Schottky (MS) devices by providing an intermediate dielectric layer. With an increase in the barrier height, the leakage current decreases while boosting the turn-on voltage with no substantial loss in the switching rate. The h-BN polycrystalline film development on a lattice-matched substrate currently necessitates a temperature over 973 K. When grown on substrates with a high lattice misfit, it results in amorphous growth [11]. According to recent studies, nanocrystalline h-BN may be produced on substrates with a significant misfit by ablation with nanosecond laser pulses and a concomitant N^2+^ plasma to force stoichiometry [12].

Thus far, the most successful method for producing pure h-BN is micromechanical cleavage, which results in a footprint of <10 μm [13]. Mechanical processing’s scale-up problems have been the key bottleneck, impeding the industrial integration of h-BN- or graphene-based products. Chemical processing approaches have recently resulted in some progress toward large-area graphene via PMMA exfoliation [14,15]. This kind of processing produces residual organic contaminants with a larger wafer-thickness fluctuation, leaving the technique unsuitable for device manufacturing. Equilibrium-based material-processing methods use a tremendously high thermal budget, which is undesirable for device feature scale-down as it facilitates increased impurity and traps diffusion. If h-BN can be locally fabricated while the rest of the device maintains near-ambient conditions, then h-BN device integration is plausible. The fabrication of layered dielectrics, such as BN with equilibrium-based approaches, faces fundamental challenges because free energy is independent of the number of layers. Such a selective materials-processing opportunity is not available with equilibrium-based approaches. Innovations in growing customized 2D h-BN and its nanoflakes for specific applications are in high demand for the development of next-generation integrated devices [16]. These limitations have prompted research into nonequilibrium ways to fabricate 2D materials at near-ambient temperatures [17,18].

The nonequilibrium procedures of nanosecond laser irradiation and zone refinement in the liquid phase were used in this study to change an a-C/a-BN heterostructure into an rGO/h-BN/Si heterostructure. The semiconductor industry is almost exclusively utilizing excimer lasers for patterning schemes. Hence, this technique can be performed on the current instrumentation, which will be cost-efficient. The process described in this study is scalable and offers a high potential for technology transfer. Notably, the limit of the resolution of the patterning is directly proportional to the wavelength of the laser used. Excimer lasers have the smallest wavelength compared with what other candidate lasers possess, and they thus possess the advantage of a higher resolution making feasible the miniaturization and complex patterning process necessary for the synthesis of 2D graphene and h-BN films [7].

This rGO/h-BN/Si heterostructure demonstrates MIS diode properties. When a-BN is exposed to temperatures higher than 2473 K and pressures greater than 7.7 GPa, it crystallizes into a cubic BN (c-BN) and h-BN phase mixture [19]. Under equilibrium-based thermal processing, both a-BN and h-BN disintegrate into boron nanoparticles and an N_2_ evolution under low-pressure conditions. Laser irradiation of a-BN melts BN, resulting in the development of h-BN on *<111>* oriented Si substrates. Because the laser irradiation process is completed in less than hundred nanoseconds, locally heating the region of interest while the rest of the device structures stay near ambient temperatures, it may be used to fabricate h-BN on wafers with pre-existing devices in a controlled manner. The identical lattice constants of graphene and h-BN along the c-axis result in the creation of films with little in-plane strain, as evidenced in the high-resolution Raman spectra, with the *G* peak in rGO located at 1570 cm^−1^ and the h-BN E_2g_ mode at 1363 cm^−1^. Such selective laser irradiation of single-crystalline h-BN films paves the way for furthering the fabrication of high-k dielectric and graphene-based devices.

## 2. Materials and Methods

### 2.1. rGO/h-BN Heterostructure Fabrication

The amorphous C/BN heterostructure was created using a pulsed laser deposition (PLD) process on *<111>* oriented Si substrates. A KrF excimer laser with a pulse width of 25 ns, a wavelength of =248 nm, and an energy density of 3.0 J/cm^2^ was used to produce 5–10 nm thick films at ambient temperature in a vacuum of 1 × 10^−6^ Torr. The KrF laser pulses were rastered across a target that is 99% pure polycrystalline h-BN with a glassy carbon sector spanning 15% of the area. Following that, the as-deposited thin films were laser-irradiated at room temperature in the air using an ArF excimer laser (6.4 eV photon energy: 20 ns) at 0.2–0.6 J/cm^2^ energy density with a spot size of 1.0 ± 0.01 cm^2^. The schematic of the experimental setup is depicted in Figure 1. To investigate the pure as-deposited film structure, the a-BN films were directly grown on a TEM grid. The laser-irradiated films were immersed in a methanol solution and ultrasonically processed. The resulting solution was drop-cast onto a thin Ted Pella lacey carbon grid #01824 (Redding, CA, USA).

### 2.2. Electrical and Structural Measurements

After forming the graphene/h-BN/Si structure, the I-V characteristics were measured using a Keithley 2400 (Cleveland, OH, USA); source meter and a current preamplifier (DL1211). An IV probe station coupled to a Keithley 4200 semiconductor analyzer was used to test the electrical properties of the manufactured devices. To analyze the Raman vibrational modes, a confocal Raman microscope system (WITex Alpha 3000 M, 532 nm source) was used. The TOF SIMS V equipment with a lateral resolution of 300 nm was utilized to identify C, O, B, and Al in multilayered C/BN heterostructures produced on Al_2_O_3_ substrates using a Cs^+^ sputtering gun. A FEI Verios 460 L was used to explore variations in the thin-film topography caused by the laser irradiation. EBSD studies were conducted with an FIB-SEM fitted with a field-emission gun to explore the phase and structure of BN during laser irradiation. Furthermore, the effects of laser irradiation on the thin-film atomic structure were studied at the atomic level using high-angle annular dark-field (HAADF) imaging and nano-diffraction studies on the aberration-corrected NION UltraSTEM (STEM, scanning transmission electron microscope).

### 2.3. Simulation Methodology

A spatiotemporal Gaussian heat source was used to study the laser irradiation. SLIM [20,21] programming was employed to understand the transformation of a-BN into h-BN. To investigate the melt-front motion toward the substrate in the direction of the maximal conduction losses, both radiative and conductive heat losses from the molten BN were integrated into the model. The thermal study for the a-BN block was carried out using a unit mesh of 1 nm. In addition to the ArF laser parameters used in the laser irradiation, the reflectance of both solid a-BN and liquid BN was used to model the laser irradiation. The thermal conductivity values for both a-BN and molten BN were estimated to be 5 and 740 W/mK, respectively [22].

## 3. Results and Discussion

The hBN/rGO/Si heterostructure was investigated first in the study, followed by the MIS diode I-V characteristics. To validate the a-BN to h-BN transformation and rGO generation by liquid-phase zone-refined growth on the selective laser irradiation, high-resolution FESEM and atomically-resolved STEM imaging, Raman and EEL spectroscopy measurements, and XRD analyses were employed. Further details on the thin-film fabrication and structural analyses are presented in the experimental section.

### 3.1. Liquid-Phase h-BN Regrowth

As polycrystalline h-BN has a large bandgap of 5.95 eV, the laser ablation of h-BN with nanosecond KrF laser pulses (photon energy: 5.0 eV) is a two-photon process. Polycrystalline h-BN ablation at low temperatures resulted in an amorphous layer deposition. To impose stoichiometry on the deposited films and maintain crystallinity, substrate temperatures >700 K are necessary under N^2+^ plasma [23]. In this study, amorphous films were melted and then quenched to produce a hexagonal boron nitride formation. The modification of the a-BN thin film by laser irradiation is shown as an SEM micrograph in Figure 2a. The thick BN films showed flaking during the ultrafast melt–regrowth due to the undulations caused by the instability in the melt front. Because the entire laser irradiation process was completed in 100 ns, the limited heat diffusion aided in the creation of sharp surfaces at the irradiation-phase boundaries. Figure 2b illustrates the high-resolution 70° tilt-corrected backscattering SEM image, showing that there was no swelling or bucking in h-BN.

In the laser-irradiated films, an EBSD analysis was used to analyze the phase change and crystal orientation. The insets in Figure 2b show the EBSD (Kikuchi) diffraction from the a-BN and laser-irradiated h-BN films. Because of the large contact volume, the Kikuchi bands associated with h-BN have a similar intensity to that of the Si substrate. When molten BN is slightly undercooled, it regrows into the stable h-BN phase. With increasing undercooling, it is possible to produce c-BN with substantial twinning. When the regrowth velocity further increases, the atoms do not have enough time to pack themselves in order, resulting in a densely packed and quenched BN (Q-BN) with local tetrahedral bonding [24]. Because of the significant amount of strain accumulated because of the lattice mismatch, the h-BN heterostructures are prone to delamination [25]. In large misfit systems, crystal growth at the melt–substrate interface occurs via domain matching epitaxy [26]. Within a few monolayers, the domain matching cancels out the lattice misfit strain, resulting in a bulk film growth with negligible crystallographic defect content and residual stresses [27,28]. As a result, nonequilibrium laser irradiation is an appropriate approach for manufacturing h-BN thin films on large-misfit substrates.

The conversion of a-BN to h-BN by the laser-annealing route represents a significant advance in the field of semiconductor fabrication. Notably, the thermal footprint of heating during laser annealing is spatially and temporally confined, down to the nanometer scale. The temperature of the film is very high and close to the melting temperature of carbon (~4000 K), resulting in zone refinement and the formation of the graphene/h-BN structure while the substrate remains at the ambient temperature. Furthermore, this technique utilizes excimer lasers. Excimer lasers have the smallest wavelength compared with other candidate lasers and thus possess the advantage of higher resolution, making feasible the miniaturization and complex patterning process for bulk manufacturing of these devices [7].

### 3.2. Scanning Transmission Electron Microscopy

Electron diffraction was used to examine the structural transition in these boron nitride films after the laser irradiation. The as-placed boron nitride films that were directly deposited on the molybdenum ring revealed a diffused, circular ring pattern, as shown in Figure 3a, confirming its short-range ordering. The structure converted into a single-crystalline h-BN after the laser irradiation, as shown in the plane-view annular dark-field picture for the h-BN monolayer in Figure 3c. Notably, the S/TEM apparatus could spatially resolve the Z-contrast of single atomic columns in boron and nitrogen. Because this was a single plane composed of triangular patterning of B-N atoms, it was directly 1:1 mapped to the real h-BN lattice. Furthermore, electron energy-loss spectroscopy was used to examine the fine structural features in boron–nitride bonding (EELS). Figure 3b compares the EEL spectrum of laser-irradiated multi- and monolayer h-BN films with different boron and nitrogen K-edge structures at 190 eV and 401 eV, respectively. These peaks correspond to the B-K and N-K edges, respectively [29,30]. Because the edge of the fine structure contains both σ* and π* orbitals, it indicates *sp^2^* hybridization for both B and N, consistent with the lattice structure observed with HAADF imaging. The spectrum was then compared with a polycrystalline, thick h-BN film plane-view specimen to investigate the changes caused by the thickness and laser irradiation procedure. In comparison with standard bulk h-BN, the spectrum showed a higher intensity for the *π** peak associated with the B-K edge, which is produced by a drop in the h-BN film’s thickness [31]. Higher core-hole effects were additionally associated with the development of B vacancies during laser irradiation could cause a modest increase in *π** intensity. Notably, no boron precipitates were found in any of these instances. This study demonstrated that laser irradiation is a feasible method for producing stoichiometric h-BN films while avoiding structural defects such as boron clusters and nitrogen vacancies.

### 3.3. Secondary Ion Mass Spectroscopy

The differences in the TOF-SIMS profiles for carbon and boron before and after laser irradiation of the multilayered a-C and a-BN films are shown in Figure 4a,b. Notably, after laser irradiation, a considerable shift in the concentration profile of C and BN was detected. Because of the zone refining, all the carbon rose to the surface, while the BN layer formed beneath it. This atomic redistribution following laser irradiation was used to determine the carbon diffusion coefficient in liquid BN. The diffusion coefficient of carbon in BN was calculated using the 2D diffusion transport equation as ∼5 × 10^−5^ cm^2^/s. Such a high diffusion coefficient is characteristic of liquid-phase diffusion [32], presenting significant evidence for subsequent carbon zone refining from molten BN.

### 3.4. Raman Spectroscopy

Two-dimensional materials such as BN and carbon have different Raman modes that can be used to distinguish between their crystalline and amorphous phases. The h-BN films exhibited high-energy phonon modes ~1364 cm^−1^, which originated from the eigenvector with *E_2g_* symmetry [33]. The Raman spectra of a-BN and laser-irradiated films are shown in Figure 5a. Notably, the ordering of basal planes in multilayer BN films could be measured using the relation: $\Gamma_{½}=141.7\;{\left(L_{a}\right)}^{-1}+8.7$, where *L_a_* is the in-plane coherence length in nm and $\Gamma_{½}$ is the FWHM of the Raman peak (cm^−1^). The bulk h-BN exhibited a full-width-half-maximum (FWHM) of 15 cm^−1^, centered at 1366 cm^−1^, resulting in a coherence length of 18.15 nm [34]. The FWHM in a-BN, on the other hand, could not be calculated due to its randomized amorphous structure. The coherence length in high-quality hBN films is in the range of 4.4–78.5 nm [35].

Irradiating ultrathin amorphous carbon films with an ArF laser resulted in the regrowth of crystalline-reduced graphene films, as shown by the *D* and *G* peaks in Figure 5b. The *G* peak was detected at ~1560 cm^−1^ due to *sp*^2^ *C-C* bond stretching, whereas the *D* peak was observed near 1350 cm^−1^ due to breathing modes resulting from crystallographic defects in the graphene basal plane. Similar to a-BN films, amorphous carbon comprises disordered *sp*^3^ and *sp*^2^ bonding states, resulting in a wide hump rather than separate vibrational modes. Figure 5c depicts the different Raman spectra obtained after the laser irradiation of a-C and a-BN/a-C multilayered films. Along with *D* and *G* vibrational modes arising from rGO, a prominent Raman peak of ~1363 cm^−1^ is shown here.

On resolving the G peak in Figure 5d, we found it arose from the rGO sitting on top of h-BN. The h-BN beneath demonstrated an FWHM of 9.7 cm^−1^ ($L_{a}$~140 nm), while the FWHM for the *G* mode of rGO was 100 cm^−1^. These low values indicated the films’ crystalline structure. In Figure 5e, the improved signal from the 2D modes (2670 cm^−1^) and D+G band at 2920 cm^−1^ also represents C-C bonding state restoration, indicating an increased GO reduction. Because the *I*_2*D*_*/I_G_* ratio is related to ordering in the C-C basal plane, the peak intensity may be used to estimate carrier mobility. Surprisingly, the reduced-GO films produced by laser irradiation had an *I*_2*D*_*/I_G_* ratio of 0.17, implying improved transport properties and greater localization lengths. Based on SIMS and Raman studies, the zone-refining of rGO from h-BN films occurs on excimer laser irradiation.

The significant difference in the solubility for BN and C in the solid and liquid phases affected zone refining. This difference arose due to the difference between the Gibbs free energy of liquid carbon −(80–120 KJ/mol) [27] and boron nitride (−228.4 KJ/mol) [36].

### 3.5. I-V Measurements

Irradiation with a nanosecond laser creates atomically sharp boundaries between the as-deposited amorphous and regrown crystalline phases. This capability can be used to develop multifunctional electronic devices. A typical Schottky diode between laser-irradiated rGO and bare n-type Si wafers is shown in Figure 6a. The I-V characteristics of this diode were analyzed by depositing an Au contact on rGO and directly placing the other probe on Si. The measurements were taken inside a cryostat in complete darkness. Because Au has a large work function, it made Ohmic contact with the rGO, but the silicon-rGO interface worked as a Schottky barrier, displaying rectifying behavior at a turn-on voltage of 0.4 V, as shown in Figure 6b.

At a low bias, direct electron tunneling [37] occurs in h-BN films, and as the bias increases, it is predicted to display nonlinear behavior driven by electron field emission, resulting in the conventional tunneling diode features. The properties of these two basic architectures were coupled to build the MIS diodes by zone-refining carbon from BN, resulting in a rGO/h-BN/Si MIS device, as illustrated in Figure 6c. The graphene layer served as a cathode for the MIS diode, while the Si served as an anode. At a 5 V bias, the MIS diode produced a current of 5 nA. The MIS diode’s threshold voltage and series resistance were 4.4 V and 1.5 × 10^7^ Ω, respectively, as shown in Figure 6d. The high threshold voltage of 4.4 V is typical of MIS devices because higher tunneling causes enhanced conduction across the dielectric (h-BN) layer. The MS diode performed better in terms of rectification characteristics and increased turn-on voltages, as predicted, whereas the MIS diode performed better in terms of the current (63 µA@5V) with similar contact areas. The leakage current between the rGO and h-BN layers was very low, implying that carriers are transported by tunneling and accumulate at the rGO/h-BN interface under the forward bias. For a better understanding of the phase change of a-BN/a-C into crystalline counterparts and their zone refinement, we used SLIM programming to model laser–solid interactions [20,21].

### 3.6. Laser-Solid-Melt Interaction Simulations

The h-BN growth from molten BN was simulated by examining the melt front’s gradual solid to liquid phase change. After laser irradiation, the melt front grew as the thermal flux traversed from the surface ad-atoms to the substrate–film interface. When the latent heat produced was insufficient to melt the underlayer, the process ended. BN melt subsequently grew back into h-BN, c-BN, or Q-BN phases based on the extent of undercooling achieved [38,39]. To better understand the mechanism underlying the evolution of h-BN film nanostructures during laser annealing, we simulated laser interactions using extremely accurate heat flow equations. *E_d_* = $\frac{K\Delta T\tau^{0.5}}{\left(1-R_{L}\right)D^{0.5}}$ denotes the threshold energy density at which melting occurs, where *K* is the thermal conductivity of h-BN, and T is the difference between the substrate temperature and the melting temperature of h-BN. The maximum melt depth achieved was calculated using Δ*x(t) = M* × (*E*(*t*) *− E_m_*)*,* where M is a material constant, and *E*(*t*) is the amount of energy absorbed by the material at time (t). Above the threshold energy density of laser irradiation, a-BN melts and may then be regrown into various phases depending on the amount of undercooling obtained [40]. For irradiating out defects in crystals, laser irradiation below the threshold energy density was used [41]. According to the simulations in Figure 7a, the threshold irradiation energy density (*E_d_*) for BN melting on Si substrate is 0.3 J/cm^2^ at temperatures over 3200 K. The models showed melt quenching rates in the 10^10^–10^11^ K/s range. Ultrafast procedures, such as nanosecond laser irradiation, are required for processing unstable materials such as a-BN, which disintegrate during equilibrium-based high-temperature thermal processing. Localized laser heating reduces undercooling ($\Delta T_{u}$) at the interface of BN melt and Si, resulting in h-BN growth. On laser irradiation at 0.3 J/cm^2^ energy density, Figure 6b shows the depth propagation of the BN melt-front with time. The Si substrate’s excellent thermal conductivity (130 W/mK) produced a steady heat flow across the melt, allowing the formation of homogenous h-BN sheets.

## 4. Conclusions

By laser irradiating a-C/a-BN films, we established a nonequilibrium-based method for fabricating rGO/h-BN/Si MIS diodes. It results in ultrashort timescale temperatures >3200 K, transforming amorphous BN to h-BN, while carbon in the a-C/a-BN multilayered structure is zone-refined toward the surface as rGO. The creation of molten C/BN and carbon zone refinement was established by the diffusion coefficient of 5 × 10^−5^ cm^2^/s. The highly textured characteristic of h-BN films was corroborated by the XRD pattern’s dominant (0002) orientation, sharp 1363 cm^−1^ Raman mode, and nano-diffraction analyses. In conditions conducive to undercooling, molten BN regrows into h-BN. The laser irradiation simulations on Si substrate indicated 0.3 J/cm^2^ as the melt-front formation threshold. The rGO/Si MS diodes had a turn-on voltage of 0.4 V, which increased to 4.4 V for the rGO/h-BN/Si MIS diodes. The diode is a tunneling diode with free charge states in graphene tunneling across the h-BN mono- and multilayer [42]. The tunneling barrier height of h-BN fabricated by a laser annealing technique on Si substrate is 3.1 eV [43]. Hence, the relevant characteristics are tunneling-driven. For laser-annealed rGO/Si diodes, the literature suggests a rGO/Si SBH from 0.11 to 0.86 eV [44,45]. Hence, a departure from Schottky diodes to MIS diodes increases the overall barrier height, providing an opportunity for significant improvement in diode rectification characteristics. In comparison with MS diodes, the I-V curves of the MIS diode demonstrated superior rectification and decreased leakage current. The ultrafast nature of laser processing inhibits the formation of crystallographic traps and defects [46], which are the primary causes of increased leakage across h-BN films grown using equilibrium-based techniques, thereby expanding the untapped potential of ultrathin devices in optoelectronics and integrated circuits useful for environmental sensing applications [47,48,49]. 

## Figures and Tables

**Figure 1 nanomaterials-12-02718-f001:**
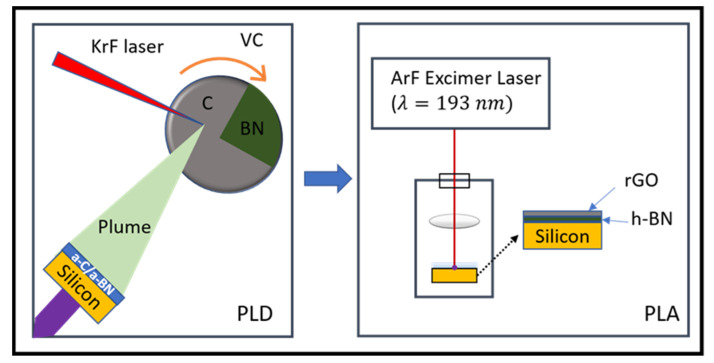
Schematic illustration of the excimer-laser-assisted synthesis of graphene/h-BN/Si MIS diode. VC, vacuum chamber.

**Figure 2 nanomaterials-12-02718-f002:**
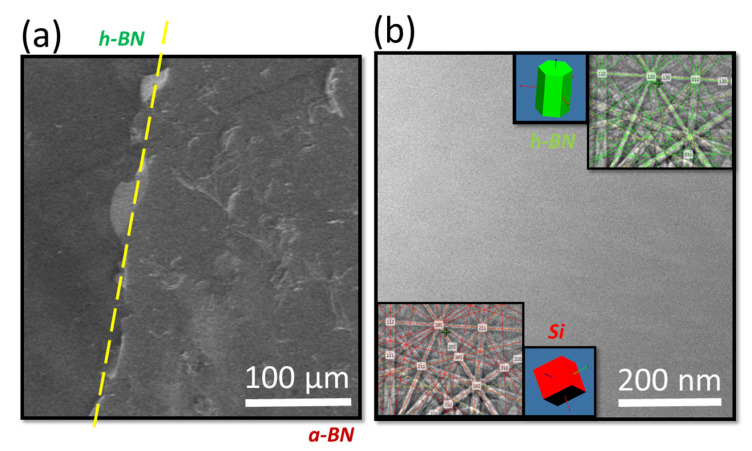
(**a**) The FESEM backscattering image of laser-irradiated h-BN/a-BN boundary; (**b**) high-resolution SEM micrograph revealing the formation of wrinkle-free high-quality h-BN films with the EBSD pattern from amorphous a-BN and laser-irradiated h-BN films, revealing the dominant Si EBSD pattern in case of a-BN acquisitions and h-BN hexagonal out-of-plane orientation in the respective insets.

**Figure 3 nanomaterials-12-02718-f003:**
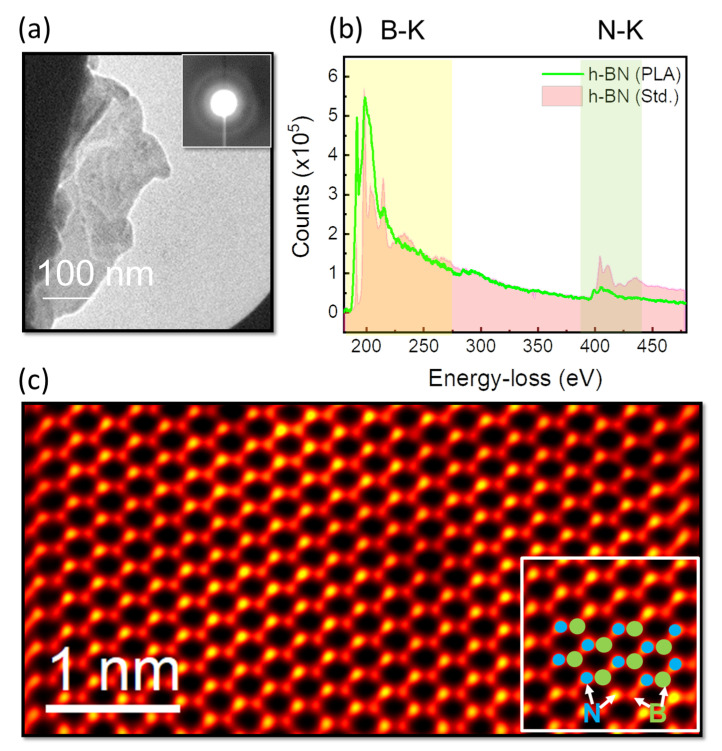
S/TEM imaging analyses: (**a**) ADF image with a nano-diffraction pattern for as-deposited a-BN; (**b**) depicts the EEL spectral acquisitions for the single-crystalline stoichiometric h-BN film. (**c**) Depicts an atomically resolved HAADF picture of a laser-irradiated h-BN monolayer, with indicated atomic positions for the h-BN lattice inset.

**Figure 4 nanomaterials-12-02718-f004:**
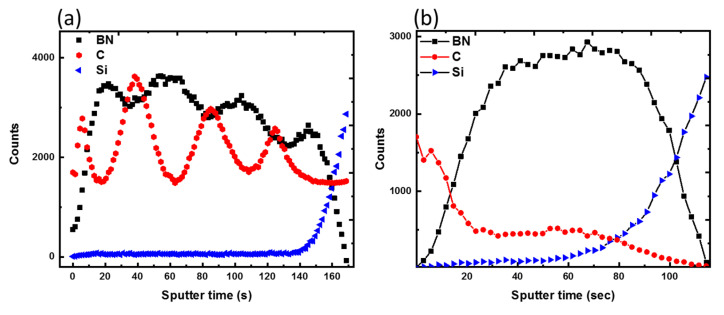
SIMS profiles of (**a**) as-deposited alternating layers of a-C and a-BN; (**b**) laser-irradiated rGO/h-BN thin films.

**Figure 5 nanomaterials-12-02718-f005:**
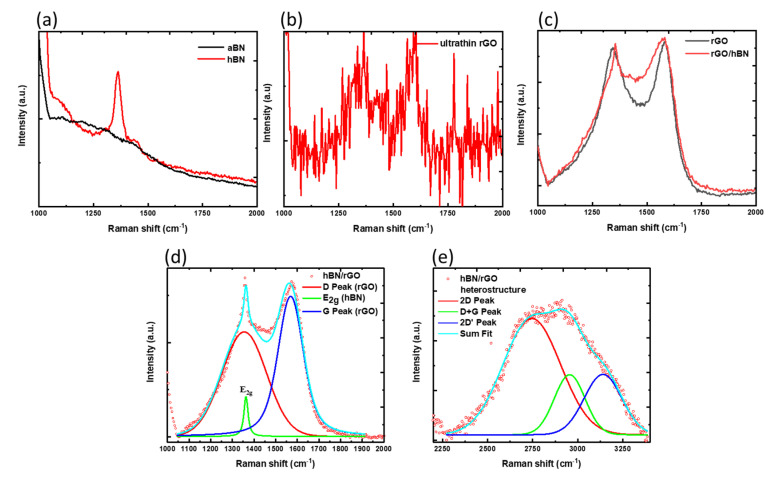
Raman spectra for (**a**) as-deposited a-BN and laser-irradiated h-BN on Si; (**b**) ultrathin amorphous carbon and (**c**) a-C/a-BN multilayered heterostructure and amorphous carbon films; (**d**) vibrational modes arising from h-BN and rGO in the laser-irradiated a-C/a-BN multilayered heterostructure, with (**e**) improved second-order (2D) peaks for rGO.

**Figure 6 nanomaterials-12-02718-f006:**
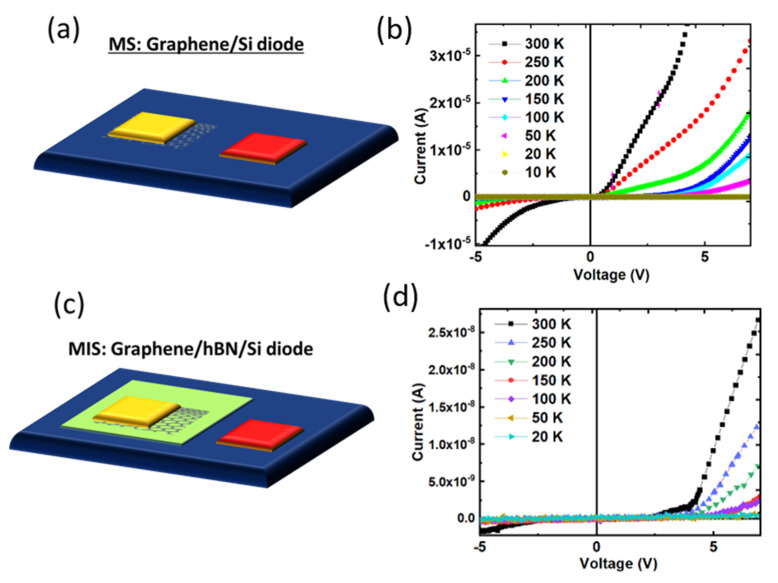
I-V measurements of h-BN films formed on laser irradiation: (**a**) Schematic and (**b**) I-V measurements from the graphene/Si MS diode formed by laser irradiation; (**c**,**d**) schematic and I-V curves, respectively, revealing the MIS diode characteristics in graphene/h-BN/Si heterostructure formed by laser irradiation.

**Figure 7 nanomaterials-12-02718-f007:**
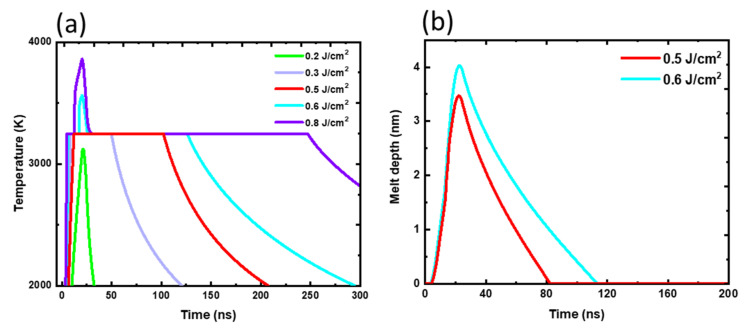
Thermal simulations of boron nitride: (**a**) temperature-time profiles for a-BN films at different laser irradiation energy densities; (**b**) melt depth as a function of time at 0.5–0.6 J/cm^2^ energy density.

## Data Availability

Not applicable.

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
