# Peer review of "Fabricating Graphene Oxide/h-BN Metal Insulator Semiconductor Diodes by Nanosecond Laser Irradiation"

_nanomaterials, 2022, doi:10.3390/nano12152718_

Round 1
Reviewer 1 Report
In "Fabricating graphene oxide/h-BN MIS diodes by nanosecond laser irradiation" the authors fabricate a graphene oxide/H-BN MIS and characterized it using Raman spectroscopy, S/TEM, and ion mass spectroscopy. The electrical properties were measured by performing an I-V characteristic.
The article is well written and the topic is interesting. Nevertheless, I have a few comments for the authors:
In the introduction there is no reference regarding graphene/Semiconductor Schottky diode, I suggest improving the bibliography by adding
doi.org/10.1016/j.physrep.2015.10.003,
10.3390/nano7070158,
doi.org/10.1088/2053-1583/aa6aa0
Also, I would like to point out that the electrical characterization has to be improved:
I suggest the authors change the Current scale in Fig 5 (b) and (d) to a logarithmic scale to highlight the MIS rectification properties.
Did the authors determinate the Schottky barrier or the Richardson constant?
Author Response
atttached as pdf

Reviewer 2 Report
In this manuscript, the authors presented a laser irradiation technique for the fabrication of rGO/h-BN heterostructure based on a-c/a-BN film. Through multiple characterizations, successful transfer of BN and C to h-BN and rGO were identified. In addition, the I-V curves and the simulation method were utilized for studying the electronic properties of the formed heterostructure and the laser irradiation effect. It is an interesting work. However, there are still some problems for this manuscript. Therefore, a major revision is suggested.
Special comments:
1. In the title, what is “MIS”? full name should be provided. In addition, the author should check the use of abbreviations in the whole text.
2. Most of the presented references are very old. The authors should focus on those references that released in the last several years.
3. What are the novelty and significance of this work? More details should be added.
4. In Part 2.1, the authors introduced the fabrication of rGO/h-BN heterostructure. However, the experimental detail is not enough. It is suggested for the authors to add a scheme on the fabrication process to make it more clear.
5. More in-depth discussion on the laser-solid melt interaction simulations should be provided.
6. What are the advantages of this laser irradiation on the fabrication of heterostructures? More discussion is needed.
Author Response
attached as pdf

Round 2
Reviewer 1 Report
I don't have any comments or suggestions to add
Reviewer 2 Report
In this revised version, the authors made great improvement according to the comments and suggestions of both referees. Now all the questions are clear and this manuscript is recommended for publication in current form.